# Microwave response of type-II superconductors at weak pinning

B. V. Pashinsky, M. V. Feigel'man,[1,2] and A. V. Andreev[3]

[1]*Floralis & LPMMC, University Grenoble-Alpes, Grenoble, France*
[2]*L. D. Landau Institute for Theoretical Physics, Chernogolovka, Russia*
[3]*Department of Physics, University of Washington, Seattle, Washington 98195, USA*
(Dated: December 13, 2022)

Theory of linear microwave response of thin films of type-II superconductors in the mixed state is developed taking into account random spatial fluctuations of the parameters of the system, such as the order parameter, diffusion coefficient, or film thickness. In the regime of collective pinning the microwave response of the system exhibits strong frequency dispersion, arising from nonequilibrium vortex core quasiparticles. The corresponding contribution to the *ac* conductivity is controlled by the inelastic relaxation time, and may exceed the usual Bardeen-Stephen conductivity. It is caused by the Debye-type inelastic relaxation. Debye mechanism of microwave losses may be responsible for strong effects of electromagnetic noise upon *dc* conductivity in the mixed state at low temperatures.

In broad range of magnetic fields, $H_{c1} < H < H_{c2}$, where $H_{c2/1}$ are the upper/lower critical fields, a type-II superconductor forms a mixed state, which hosts Abrikosov vortices. The superfluid and dissipative properties the system are closely related to the pinning and dynamics of the vortices. For a thin film of type-II superconductor in a magnetic field normal to the sample, which is subjected to an in-plane *ac* electric field $\boldsymbol{E}(t) = \mathrm{Re}\boldsymbol{E}_\omega e^{-i\omega t}$, the induced density of macroscopic transport current $\boldsymbol{j}(t) = \mathrm{Re}\boldsymbol{j}_\omega e^{-i\omega t}$ may be written for sufficiently low frequencies as

$$\boldsymbol{j}_\omega = \left[\frac{ic^2}{4\pi\omega\lambda_{\mathrm{eff}}^2(H)} + \sigma\right]\boldsymbol{E}_\omega. \tag{1}$$

Here $\sigma$ is the conductivity and $\lambda_{\mathrm{eff}}(H)$ is the effective penetration depth of the magnetic field. The latter is very sensitive to the character of vortex pinning. In particular, for $H \ll H_{c2}$, the effective penetration depth $\lambda_{\mathrm{eff}}(H)$ is given by the Campbell relation [1]

$$\lambda_{\mathrm{eff}}^2(H) = \lambda_L^2 + \frac{\Phi_0 H d}{8\pi k(0)}. \tag{2}$$

Here the first term is inversely proportional to the superfluid density $N_s$ and is expressed in terms of the London penetration length $\lambda_L = \sqrt{mc^2/4\pi N_s e^2}$. In the second term $\Phi_0 = \pi\hbar c/|e|$ is the flux quantum, $d$ is the film thickness, and $k(0)$ is the "spring constant" of vortex pinning. Equation (2) reflects the fact that the motion of vortices creates dissipation, and superfluid response is possible only in the presence of pinning, i.e. when $k(0) > 0$.

The dissipative conductivity $\sigma$ satisfies the Bardeen-Stephen relation[2], and has been analyzed in different regimes [3–6],

$$\sigma_{BS} = \zeta\sigma_n\frac{H_{c_2}}{H}. \tag{3}$$

Here $\zeta \sim 1$ depends on the details of the system, $\sigma_n$ is the conductivity in normal metal $\sigma_n = e^2\nu_n D_n$, where $D_n$ and $\nu_n$ are the electron diffusion coefficient and the density of states at the Fermi energy, respectively. We note that the Bardeen-Stephen conductivity is insensitive to the details of pinning, and is of the same order as the conductivity in the flux-flow regime. It arises from elastic scattering of vortex core quasiparticles and is proportional to the elastic relaxation time $\tau_{el}$.

A different dissipation mechanism was shown to exist in superconductors [7–11] when some vector breaking the time-reversal symmetry, e.g. a *dc* current, is present in the system. This mechanism is similar to the Debye absorption mechanism and is caused by inelastic scattering of quasiparticles. When present, it provides a contribution to the conductivity, $\sigma_{DB}$, which is proportional to the inelastic relaxation time $\tau_{in}$. Since typically $\tau_{in} \gg \tau_{el}$, the Debye contribution $\sigma_{DB}$ may exceed the conventional conductivity.

The Debye absorption mechanism arises in superconductors because the quasiparticle density of states $\nu(\epsilon)$ depends on the superfluid momentum $\boldsymbol{p}_s = -i\hbar\boldsymbol{\nabla}\chi - \frac{2e}{c}\boldsymbol{A}$, where $\chi$ is the phase of the order parameter, and $\boldsymbol{A}$ is the vector potential. In the presence of an electric field $\boldsymbol{E}(t) = -\dot{\boldsymbol{A}}/c$ the superfluid momentum acquires a time-dependent correction $\delta\boldsymbol{p}_s$, which satisfies the condensate acceleration equation $\frac{d}{dt}\delta\boldsymbol{p}_s = 2e\boldsymbol{E}(t)$. The linear dependence of the quasiparticle density of states $\nu(\epsilon)$ on $\delta\boldsymbol{p}_s$ requires the presence of another vector which breaks time reversal symmetry. As a result the Debye contribution to conductivity discussed in previous studies arose either in the nonlinear regime or in the presence of a *dc* supercurrent [7–10], or in two-dimensional samples of non-centrosymmetric superconductors in the presence of an in-plane magnetic field [11]. In either case, the Debye contribution to differential conductivity is anisotropic; it is present only for the electric field along the symmetry breaking vector.

In this Letter we show that in the mixed state of type-II superconductors the Debye contribution to the conductivity arises in the linear response regime in the absence of a *dc* supercurrent, and is isotropic in the plane perpendicular to the magnetic field. The explanation for

this somewhat unexpected result is that microwave absorption is caused by vortex core quasiparticles. While the time reversal symmetry is obviously broken, the local vector necessary for the linear Debye absorption arises from the pinning force on the vortex. Since the direction of the pinning force is random in space the macroscopic conductivity is isotropic. Thus, in contrast to the traditional picture, the *ac* dissipative conductivity in this case is closely related to pinning properties. Since the Debye contribution to the conductivity exhibits strong frequency dispersion at $\omega \sim 1/\tau_{in}$ the effective penetration depth, describing the superfluid response also acquires strong frequency dependence. In particular, for weak pinning, $\lambda_{\text{eff}}$ at high frequencies can be much smaller than the static value in Eq. (2).

The motivation for this study originates partly from the recent observation [12] that rather weak electromagnetic noise in the microwave frequency range may change drastically the low-temperature behavior of disordered superconductors in the mixed state. This indicates that low-temperature dissipation in the mixed state of superconductors may be much larger than expected from Eq.(3); the Debye mechanism we study here might be responsible for the effect.

Below we consider thin superconducting films whose thickness $d$ is small as compared to the skin depth, so that the electric field $\boldsymbol{E}(t)$ and the density of induced transport current, $\boldsymbol{j}(t)$ are nearly spatially uniform. The vortices oscillate under the action of the Lorentz force exerted by the current $\boldsymbol{j}(t)$ and create non-equilibrium quasi-particle distribution inside vortex cores, leading to energy dissipation.

Depending on the magnitude of the individual pinning force, there can be two fundamentally different cases: pinning of individual vortices and collective pinning - respectively strong and weak. We will analyse the case of collective pinning. In this case, the positional order of the vortex lattice is preserved within a domain of size $L_{cor} \gg \sqrt{\Phi_0/H}$ but the long range order is destroyed by collective pinning [4, 13, 14]. Another length scale, $L_p$, usually referred to as the Larkin length, characterizes the size of vortex domains that are collectively pinned by disorder. At $H \lesssim H_{c2}$ the two length scales are of the same order of magnitude, while at low fields, $H \ll H_{c2}$, the Larkin length $L_p$ is shorter than $L_{cor}$; it is $L_p$ which is important for our purposes below. Collective pinning scenario corresponds to the situation of $L_p \gg \sqrt{\Phi_0/H}$.

In the presence of macroscopic transport current $\boldsymbol{j}$ through the sample the vortices are displaced from their equilibrium positions. The displacement $\boldsymbol{u}_a$ of the $a$-th vortex is determined by the balance between the Lorentz force and the pinning force $\boldsymbol{f}_a$. In a steady state, the pinning force can be obtained by differentiating the free energy of the system with respect to the vortex displacement, $\boldsymbol{f}_a = -\frac{dF\{\boldsymbol{u}\}}{\boldsymbol{u}_a}$. Due to elastic stresses in the vortex lattice both the free energy $F\{\boldsymbol{u}\}$ and the pinning force

$\boldsymbol{f}_a$ depend on the displacements of all the vortices, resulting in a collective character of pinning.

A particularly interesting contribution to the pinning force arises from the vortex core quasiparticles. It is caused by the dependence of the quasiparticle energy levels $\epsilon_{a,j}(\boldsymbol{u}_a)$ (where $j$ labels the energy level, and $a$ - the vortex) on the vortex displacements $\boldsymbol{u}_a$. This dependence originates from the spatial variations gap $\Delta(\boldsymbol{r}) = \Delta_0 + \delta\Delta(\boldsymbol{r})$, diffusion coefficient $D(\boldsymbol{r}) = D_0 + \delta D(\boldsymbol{r})$, and the film thickness $d(\boldsymbol{r})$. We use a shorthand notation $\alpha(\boldsymbol{r})$ for these quantities, and characterize their spatial variations by the correlation function

$$\langle \delta\alpha\left(\boldsymbol{r}_1\right) \delta\alpha\left(\boldsymbol{r}_2\right)\rangle = \left\langle (\delta\alpha)^2 \right\rangle g\left(\frac{|\boldsymbol{r}_1 - \boldsymbol{r}_2|}{r_c}\right), \quad (4)$$

where $r_c$ is the correlation radius, $g(r/r_c)$ is normalized to $g(0) = 1$.

In the presence of spatial variations of the parameters $\alpha(\boldsymbol{r})$, the displacement of the vortices caused by the microwave field causes time-dependence of the quasiparticle energy levels in the vortex cores, creating a nonequilibrium population of these levels. In this case the pinning force depends not only on the instantaneous positions of the vortices, but also on the distribution function of the vortex core quasiparticles. As a result it acquires a strong temporal dispersion on the time scale of order of the inelastic relaxation time $\tau_{in}$. To linear order in the displacements the pinning force is given by

$$\boldsymbol{f}_a(t) = -\frac{dF\{\boldsymbol{u}\}}{d\boldsymbol{u}_a} - \sum_j \delta n_{a,j}(t) \boldsymbol{\nabla}_{\boldsymbol{u}_a}\epsilon_{a,j} - \eta\dot{\boldsymbol{u}}_a. \quad (5)$$

Here $\eta = \zeta\Phi_0 dH_{c2}\sigma_n/2c^2$, with $d$ being the film thickness and $\Phi_0 = \pi\hbar c/|e|$ - the flux quantum, $\zeta$ is the friction coefficient, which corresponds to the Bardeen-Stephen expression for the conductivity. Microscopically, the last term in the RHS of (5) originates from non-adiabatic transitions between quasi-particle electron-hole states localized in the vortex core. The second term in the RHS describes the contribution of nonequilibrium vortex core quasi-particles to the pinning force. It is expressed in terms of the nonequilibrium occupancy of the quasi-particle levels, $\epsilon_{a,j}(\boldsymbol{u}_a)$, $\delta n_{a,j}(t) = n_a(\epsilon_{a,i}, t) - n_F(\epsilon_{a,j}(\boldsymbol{u}_a))$, with $n_F(\epsilon)$ being the Fermi distribution, and the energy level sensitivies with respect to $\boldsymbol{u}_a$ at zero displacements, $\boldsymbol{\nabla}_{\boldsymbol{u}_a}\epsilon_{a,j} \equiv \frac{d}{d\boldsymbol{u}_a}\epsilon_{a,j}(\boldsymbol{u}_a)\big|_{\boldsymbol{u}_a=0}$. This term may be obtained by evaluating the rate of change of the energy of the quasiparticles, $\boldsymbol{f}_a^{qp} \cdot \dot{\boldsymbol{u}}_a = -\frac{dE^{qp}}{dt}$, and applying Ehrenfest's theorem,

$$\frac{dE_a^{qp}}{dt} = \left\langle \partial_t \hat{H}_a^{qp} \right\rangle = \sum_j n_{a,j}(t)\dot{\boldsymbol{u}}_a \cdot \boldsymbol{\nabla}_{\boldsymbol{u}_a}\epsilon_{a,j}.$$

Here $\hat{H}_a^{qp}$ is the Hamiltonian of the vortex core quasiparticles and $\langle\ldots\rangle$ denotes averaging over the quasiparticle distribution function.

Below we focus on the regime of weak collective pinning [13]. In this case, due to the stiffness of the vortex lattice, the displacements of the vortices turn out to be nearly uniform within the Larkin domain. The average displacement $\bar{\boldsymbol{u}}(t)$ is determined by balancing the spatial average of the pining force (5) with the Lorentz force density. The Lorentz force per unit area is given by

$$\boldsymbol{f}_L = \frac{d}{c}[\boldsymbol{j} \times \boldsymbol{H}], \qquad (6)$$

where $d$ is the film thickness, and $\boldsymbol{j}(t)$ is the density of macroscopic transport current. In the presence of microwave radiation the time-dependent part of the transport current depends on both the microwave field and the displacement of the vortex lattice, and can be expressed as,

$$\boldsymbol{j}(t) = -\frac{e^2 N_s}{mc}\left(\delta\boldsymbol{A}(t) + \frac{1}{2}[\boldsymbol{H} \times \bar{\boldsymbol{u}}(t)]\right), \qquad (7)$$

where $N_s$ is the macroscopic superfluid density in the absence of vortices, $\delta\boldsymbol{A}(t)$ is a uniform time-dependent vector potential, which is related to the microwave electric field by $\boldsymbol{E}(t) = -\delta\dot{\boldsymbol{A}}(t)/c$. The expression in Eq. (7) follows from invariance under magnetic translations. For example, in the symmetric gauge, a magnetic translation by $\boldsymbol{u}$ transforms the order parameter and the vector potential as: $\psi(\boldsymbol{r}) \rightarrow \hat{T}_a\psi(\boldsymbol{r}) = \psi(\boldsymbol{r}+\boldsymbol{u})e^{i\frac{|e|}{\hbar c}\boldsymbol{r}\cdot[\boldsymbol{H}\times\boldsymbol{u}]}$, $\boldsymbol{A}(\boldsymbol{r}) \rightarrow \boldsymbol{A}(\boldsymbol{r}+\boldsymbol{u}) - \frac{1}{2}[\boldsymbol{H} \times \boldsymbol{u}]$. Since the vortex lattice is displaced by $\boldsymbol{u}$ the expression in Eq. (7) remains invariant.

Averaging the pinning force (5) over space and equating it to the negative of the Lorentz force density, and using Eqs. (6), and (7), we get

$$k(0)\bar{\boldsymbol{u}}(t) + \eta\,\dot{\bar{\boldsymbol{u}}}(t) = \frac{\Phi_0 d}{8\pi H \lambda_L^2}\left(2[\boldsymbol{H} \times \delta\boldsymbol{A}(t)] - H^2\bar{\boldsymbol{u}}(t)\right)$$
$$- \sum_j \overline{\delta n_{a,j}(t)\boldsymbol{\nabla}_{\boldsymbol{u}_a}\epsilon_{a,j}}. \qquad (8)$$

We denoted averaging over the vortices by the overline, $\overline{\cdots}$ and expressed the average static pinning force per vortex in terms of a "spring constant" $k(0)$ as $\frac{dF\{\boldsymbol{u}\}}{d\boldsymbol{u}_a} \approx k(0)\bar{\boldsymbol{u}}$.

In the static limit $\bar{\boldsymbol{u}}(t)$ and the non-equilibrium part of the quasi-particle distribution function vanish. In this case substituting Eq. (8) into Eq. (7) we obtain $\boldsymbol{j} = -\frac{c}{4\pi\lambda_{\text{eff}}^2(H)}\delta\boldsymbol{A}$, where the effective penetration depth in the pinned vortex state, $\lambda_{\text{eff}}(H)$, is given by the Campbell relation [1], see Eq.(2). At relatively weak pinning the spring constant $k(0)$ is small, and the dominant contribution to $\lambda_{\text{eff}}$ is provided by the second term in (2); we will refer to such a situation as the Campbell limit. In this case, since $k(0)$ is independent of $H$ for $H \ll H_{c2}$, we have $\lambda_{\text{eff}}(H) \propto \sqrt{H}$.

For a general time-dependent microwave field the linear relation between $\bar{u}(t)$ and $\delta\boldsymbol{A}(t)$ becomes nonlocal in time. Its determination requires solving the evolution equation for the quasiparticle distribution function $\delta n_a(t)$. To linear order in $\bar{\boldsymbol{u}}(t)$ the evolution equation for the distribution function of vortex core quasiparticles has the form [8–10]

$$\left(\frac{d}{dt} + \frac{1}{\tau_{in}}\right)\delta n_{a,j}(t) = -\dot{\bar{\boldsymbol{u}}}(t) \cdot \boldsymbol{\nabla}_{\boldsymbol{u}_a}\epsilon_{a,j}\frac{dn_F(\epsilon_{a,j})}{d\epsilon_{a,j}}, \qquad (9)$$

Let us consider a superconducting film subjected to a monochromatic microwave field $\delta\boldsymbol{A}(t) = \text{Re}\left(\boldsymbol{A}_\omega e^{-i\omega t}\right)$. Introducing complex amplitudes for all time-dependent quantities, e.g. $\bar{\boldsymbol{u}}(t) = \text{Re}\left(\boldsymbol{u}_\omega e^{-i\omega t}\right)$, and $\delta n_{a,j}(t) = \text{Re}\left(\delta n_{a,j}(\omega)e^{-i\omega t}\right)$ we get from Eq. (9)

$$\delta n_{a,j}(\omega) = \frac{\omega\,\boldsymbol{u}_\omega \cdot \boldsymbol{\nabla}_{\boldsymbol{u}_a}\epsilon_{a,j}}{\omega + \frac{i}{\tau_{in}}}\left(-\frac{dn_F(\epsilon_{a,j})}{d\epsilon_{a,j}}\right).$$

Substituting this expression into Eq. (8) yields

$$\left(k(\omega) + \frac{\Phi_0 H d}{8\pi\lambda_L^2}\right)\boldsymbol{u}_\omega = \frac{\Phi_0 d}{4\pi H \lambda_L^2}[\boldsymbol{H} \times \delta\boldsymbol{A}_\omega], \qquad (10)$$

where $k(\omega) = \tilde{k}(\omega) - i\omega\eta$ is determined via another function $\tilde{k}(\omega)$:

$$\tilde{k}(\omega) = k(0) - \frac{i\omega\tau_{in}\delta k}{1 - i\omega\tau_{in}}. \qquad (11)$$

The quantity $\delta k = \tilde{k}(\infty) - k(0)$ has the meaning of the change in the pinning "spring constant" between the high frequency limit, $\omega\tau_{in} \gg 1$, and the static limit $\omega\tau_{in} \ll 1$. It is expressed in terms of the sensitivities of the quasiparticle levels to the displacement of the vortex as

$$\delta k = \frac{1}{2}\sum_j \overline{(\boldsymbol{\nabla}_{\boldsymbol{u}_a}\epsilon_{a,j})^2\left(-\frac{dn_F(\epsilon_{a,j})}{d\epsilon_{a,j}}\right)}. \qquad (12)$$

To obtain this expression we rely on the isotropy of gradients $\boldsymbol{\nabla}_{\boldsymbol{u}_a}\epsilon_{a,j}$ in a random sample. Note that frequency dependence of $\tilde{k}(\omega)$ is expressed in Eq. (11) in terms of three parameters; the inelastic relaxation time $\tau_{in}$, and the static spring constant $k(0)$, and its change $\delta k$.

Substituting Eqs. (11) and (10) into Eq. (7) we get the following expression for the $ac$ conductivity,

$$\sigma(\omega) = \frac{ic^2}{4\pi\omega}\frac{1}{\lambda_L^2 + \frac{\Phi_0 H d}{8\pi k(\omega)}}. \qquad (13)$$

In the Campbell limit, $\lambda_{\text{eff}} \gg \lambda_L$, we may neglect $\lambda_L^2$ in the denominator. In this case the real part of conductivity is given by

$$\text{Re}\,\sigma(\omega) = \frac{2c^2\tau_{in}\delta k}{\Phi_0 dH(1 + \omega^2\tau_{in}^2)} + \frac{H_{c2}}{H}\zeta\sigma_n. \qquad (14)$$

Equation (14) is our main result. It shows that apart from the Bardeen-Stephen contribution, the real part of

part of the *ac* conductivity of a pinned vortex lattice contains an additional contribution caused by inelastic scattering of quasiparticles. This contribution arises from Debye relaxation mechanism and exhibits a strong dispersion at frequency scales on the order of inelastic relaxation rate $1/\tau_{in}$. At low frequencies this contribution is proportional to $\tau_{in}$. Since $\tau_{in}$ may exceed the elastic relaxation time by several orders of magnitude, at low frequencies the Debye contribution may exceed the Bardeen-Stephen contribution. The expression in Eq. (14) applies at frequencies below the elastic relaxation rate $1/\tau_{el}$, and describes the frequency dispersion of the *ac* conductivity in terms of the inelastic relaxation rate $1/\tau_{in}$, and $\delta k$. The latter is defined by Eq. (12) and corresponds to the quasiparticle contribution to the "spring constant" of the pinning force. In accordance with the Le Chatelier's principle $\delta k > 0$. Under weak pinning conditions $L_p \gg \xi$, the "non-equilibrium" contribution to the spring constant may appear to be dominant, $\delta k \gg k(0)$. If the inequality $\delta k \gg \eta/\tau_{in}$ is fulfilled, in the intermediate frequency region $1/\tau_{in} \ll \omega \ll \delta k/\eta$ the entire response will be mainly reactive, with effective penetration depth

$$\lambda_{eff}^2(H;\infty) = \lambda_L^2 + \frac{\Phi_0 H d}{8\pi(k(0)+\delta k)}, \qquad (15)$$

which may differ considerably from the zero-frequency value given by Eq.(2).

Equation (12) expresses $\delta k$ in terms of the sensitivities of individual energy levels of vortex-core quasiparticles to the vortex displacement. Since in practically all situations the mean level spacing in the vortex core is negligibly small, it is convenient describe the vortex core quasiparticles by a continuum density of states $\nu_a(\epsilon, \boldsymbol{u})$. The transformation from the quasiparticle energy levels to the density of states is equivalent to the transformation between Lagrangian and Eulerian variables in the hydrodynamics of one-dimensional liquids. The formulas in the Eulerian representation can be obtained [8, 9] by replacing in Eq. (12) the summation over levels by the integration, $\sum_j \ldots \to \int_0^\infty \ldots \nu_a(\epsilon, \boldsymbol{u}) d\epsilon$, and the level sensitivity as $\boldsymbol{\nabla}_{\boldsymbol{u}_a}\epsilon_{a,j} \to \boldsymbol{V}_a(\epsilon)$ where $\boldsymbol{V}_a(\epsilon)$ is expressed in terms of the density of states $\nu_a(\epsilon, \bar{\boldsymbol{u}})$ of vortex core quasiparticles in the form

$$\boldsymbol{V}_a(\epsilon) = -\frac{1}{\nu_a(\epsilon,0)} \int_0^\epsilon d\epsilon \left.\frac{\partial \nu_a(\epsilon,\bar{\boldsymbol{u}})}{\partial \bar{\boldsymbol{u}}}\right|_{\bar{\boldsymbol{u}}=0}. \qquad (16)$$

Making these substitutions in Eq. (12) we obtain the following expression

$$\delta k = -\frac{1}{2} \int_0^\infty d\epsilon \frac{dn_F(\epsilon)}{d\epsilon} \overline{\nu_a(\epsilon)\boldsymbol{V}_a(\epsilon)^2}. \qquad (17)$$

Let us now estimate the electron contribution to the spring constant $\delta k$ in thin films. The density of states

of the vortex core quasiparticles may be estimated as $\nu(\epsilon) \sim \nu_0 \xi^2(\boldsymbol{r}) d(\boldsymbol{r})$, where $\nu_0 = mp_F/(2\pi^2\hbar^3)$ is the normal state density of states per single spin projection, and $d(\boldsymbol{r})$ and $\xi(\boldsymbol{r})$ are, respectively, the local film thickness and coherence length. Therefore the level sensitivity in Eq. (16) may be estimated as $\boldsymbol{V}_a(\epsilon) \sim \epsilon\boldsymbol{\nabla} \ln \left[\xi^2(\boldsymbol{r})d(\boldsymbol{r})\right]$. Substituting this into (17) yields

$$\delta k \sim T^2 \nu_0 \xi^2 d \left\langle \left(\boldsymbol{\nabla} \ln \left[\xi^2(\boldsymbol{r})d(\boldsymbol{r})\right]\right)^2 \right\rangle, \qquad (18)$$

where $T$ is the temperature. In obtaining this estimate we tacitly assumed that energy relaxation is caused by electron-phonon scattering. The reason is that for energy-independent density of quasiparticle states in the vortex core, the nonequilibrium distribution of quasiparticles generated by the vortex displacement corresponds to a change of the quasiparticle temperature. Its relaxation requires transfer of energy between the quasiparticles and phonons. Using Eqs. (18) and (4) we get the following estimate

$$\delta k \sim \frac{T^2 \nu_0 \xi^2 d}{r_c^2} \frac{\left\langle (\delta\alpha)^2 \right\rangle}{\langle\alpha\rangle^2}.$$

Substituting this into Eq. (14), at $\omega\tau_{in} \ll 1$ we estimate the ratio of the Debye contribution to the conductivity, $\sigma_{DB}$, to the Bardeen-Stephen result, $\sigma_{BS}$, as

$$\frac{\sigma_{DB}}{\sigma_{BS}} \sim \frac{\tau_{in}}{\tau_{el}} \frac{T^2\xi^2}{\hbar^2 v_F^2} \frac{\xi^2}{r_c^2} \frac{\left\langle (\delta\alpha)^2 \right\rangle}{\langle\alpha\rangle^2}. \qquad (19)$$

This estimate for $\sigma_{DB}$ in the linear response regime turns out to be of the same order as that in the flux flow regime [10].

If variations of the film thickness, $\delta d(\boldsymbol{r})$, are the leading source of disorder then in the above expressions one should set $\left\langle (\delta\alpha)^2 \right\rangle/\langle\alpha\rangle^2 \to \langle(\delta d)^2\rangle/d^2$. In the dirty case, $\xi^2 \sim \hbar v_F^2 \tau_{el}/\Delta$, we get

$$\frac{\sigma_{DB}}{\sigma_{BS}} \sim \frac{v_F^2 \tau_{in}\tau_{el}}{r_c^2} \frac{T^2}{\Delta^2} \frac{\langle(\delta d)^2\rangle}{d^2}.$$

In the clean regime, where the elastic mean free path is limited by surface scattering, $\xi^2 \sim \hbar v_F d/\Delta$, and $\tau_{el} \sim d/v_F$. We thus get

$$\frac{\sigma_{DB}}{\sigma_{BS}} \sim \frac{v_F \tau_{in} d}{r_c^2} \frac{T^2}{\Delta^2} \frac{\langle(\delta d)^2\rangle}{d^2}.$$

In conclusion, we have shown that the Debye mechanism of relaxation may lead, for a mixed state of type-II superconductors, to a strong enhancement of *ac* conductivity $\sigma(\omega)$ at relatively low frequencies, $\omega \leq 1/\tau_{in}$. For the case of thin films the corresponding result is provided by Eq. (14), assuming the Campbell limit, $\lambda_{\text{eff}}(H) \gg \lambda_L$, is realized. The relative importance of the Debye contribution to conductivity, $\sigma_{DB}$, and the Bardeen-Stephen

contribution, $\sigma_{BS}$, is characterized by the estimate. The linear response conductivity $\sigma_{DB}$ turns out to be of the same order as the Debye contribution to the conductivity in the flux flow regime obtained in Ref. 10. The Debye contribution can be dominant at low temperatures $T \ll T_c$ where inelastic relaxation time $\tau_{in}$ strongly increases and can exceed $\tau_{el}$ by several orders of magnitude. The new microwave absorption mechanism we found may be responsible for anomalously strong effects of weak electromagnetic noise on $dc$ transport, which was reported in Ref. [12].

Due to the presence of the Debye relaxation mechanism the reactive part of the response may demonstrate strong frequency dispersion if $\delta k \geq k(0)$, see Eqs.(2) and (15). Such a situation should arise naturally in the case of weak pinning with the Larkin length $L_p \gg \xi$. Indeed, the equilibrium "spring constant" $k(0)$ scales as a negative power of $L_p$ due to collective nature of pinning, while the non-equilibrium part $\delta k$ in Eq. (17) depends on the local properties of disorder near each separate vortex and does not depend on $L_p$.

We note that in recent experiments on microwave absorption in a mixed state of $d$-wave superconductors strong frequency dispersion of microwave impedance at rather low frequencies was observed [15]. Although in $d$-wave materials in addition to vortex core quasiparticles, a nodal quasi-particles away from vortex cores are present, the mechanism of strong low frequency dispersion may be similar to the one discussed here. The detailed study of microwave absorption in $d$-wave materials is outside the scope of the present study.

We are grateful to Boris Spivak and David Broun for discussions. This work was supported, in part, by the US National Science Foundation through the MRSEC Grant No. DMR-1719797, the Thouless Institute for Quantum Matter, and the College of Arts and Sciences at the University of Washington (A.V.A.), and by the RSF Grant No. 20-12-00361 (B.V.P. and M.V.F.).

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
