# Peer review of "Microwave response of type-II superconductors at weak pinning"

_SciPost Physics_

## Round 1 · Referee Report · Aleksandr Svetogorov (Referee 1) · 2023-1-3

Strengths

1 - Relevant topic
2 - New effect predicted
3 - Detailed analytical analyses
4 - Comprehensive

Report

In their work the authors study a microwave response of thin superconducting films in the mixed state (type-II superconductors). More specifically they consider the case of random fluctuations in the system parameters and weak pinning regime. The authors show that on top of the previously studied Bardeen-Stephen contribution to the ac conductivity a new term arises due to nonequilibrium vortex core quasiparticles. The later can be related to the Debye absorption mechanism, however, in contrast to the typical case when it is present only in case of anisotropy (and the contribution to the ac conductivity is normally anisotropic as well) the mechanism described in the manuscript is macroscopically isotropic (local anisotropy for the quasiparticles in the vortex core is determined by the pinning force, which is random). The authors estimate the contribution of this mechanism to the ac conductivity and show that it can be dominating and leading to a strong enhancement of the ac conductivity at low frequencies. Two experimental results showing some unexpected frequency dispersion of microwave impedance are briefly discussed in the context of this newly discovered mechanism.

The manuscript is coherent and comprehensive, the effect discussed in the text is important for experiments with thin superconducting films with vortices subjected to microwave radiation (also the unwanted external one) as well as of fundamental significance (macroscopically isotropic Debye absorption). From my side I can only suggest adding a brief discussion about the ways to experimentally verify the effect (i.e. for me it is not evident if it is feasible to distinguish the contribution by measuring the ac conductivity at different temperatures as the inelastic scattering time also depends on the temperature). In general I recommend the manuscript for publication in SciPost Physics.

Requested changes

1 - A brief discussion of possible ways to detect the effect experimentally

---

## Round 1 · Referee Report · Anonymous (Referee 2) · 2023-1-15

Report

Thin superconducting films with collectively pinned Abrikosov vortices are considered. The authors obtained an interesting low-frequency contribution to ac-conductivity. The mechanism of the effect is similar to the Debye contribution to the conductivity, but it is isotopic in the plane of the film. The isotropy of the effect arises from the averaging of the random pinning forces. The result diverges at low frequencies $\sim \tau_{in}$.
The work looks interesting and relevant, undoubtedly deserving of publication. But as a reader I would like the authors to explain which factor restricts the low-frequency divergence at $\tau_{in} \to \infty$.

---

## Round 1 · Referee Report · Anonymous (Referee 3) · 2023-1-19

Strengths

  • New proposal for dissipation mechanism in vortex state

Weaknesses

  • Limitations in the discussion of previous works in the field

Report

The manuscript describes a contribution to microwave conductivity from motion of energy levels in the core of pinned vortices.

Similar dissipation mechanism manifests also in other superconductor systems, with certain cases considered also by some of the authors in previous works. To my knowledge, the specific mechanism studied here is new, and is a nontrivial extension.

The calculations are presented clearly, and the results are physically plausible.

The relation to the existing (experimental) work on ac response in vortex state could perhaps use additional comments in the manuscript. Although tau_in >> tau_el could mean the mechanism here is fairly visible, apparently its presence is not readily identified from existing experiments, as the authors only cite two recent papers which might be related. In light of the estimates here, is this as expected? Some additional discussion of this point could be useful for readers.

The problem studied is interesting, and I recommend publication in SciPost Physics after considering the comments.

---

## Editorial Decision

resubmitted